# Enhancing Reasoning through Process Supervision with Monte Carlo Tree Search

**Shuangtao Li[1], Shuaihao Dong[1], Kexin Luan[1], Xinhan Di[1], Chaofan Ding[1],**

[1]AI Lab, Giant Network

lishuangtao@ztgame.com, dongshuaihao@ztgame.com, luankexin@ztgame.com, dixinhan@ztgame.com,
dingchaofan@ztgame.com

## Abstract

Large language models (LLMs) have demonstrated their remarkable capacity across a variety of tasks. However, reasoning remains a challenge for LLMs. To improve LLMs' reasoning ability, process supervision has proven to be better than outcome supervision. In this work, we study using Monte Carlo Tree Search (MCTS) to generate process supervision data with LLMs themselves for training them. We sample reasoning steps with an LLM and assign each step a score that captures its "relative correctness," and the LLM is then trained by minimizing weighted log-likelihood of generating the reasoning steps. This generate-then-train process is repeated iteratively until convergence. Our experimental results demonstrate that the proposed methods considerably improve the performance of LLMs on two mathematical reasoning datasets. Furthermore, models trained on one dataset also exhibit improved performance on the other, showing the transferability of the enhanced reasoning ability.

## Introduction

Although today's large language models (LLMs) can perform excellently on a variety of language tasks (Zhao et al. 2023), even approaching human levels, reasoning remains an unaddressed challenge for them (Huang and Chang 2023; Valmeekam et al. 2022). To enhance their reasoning ability, some studies use Chain-of-Thought (CoT) prompting (Nye et al. 2021; Wei et al. 2022; Kojima et al. 2022) to encourage LLMs to decompose given problems and think step by step, but the elicited reasoning ability is still limited. Nevertheless, these works emphasize the importance of step-by-step reasoning for LLMs.

To enhance LLMs' reasoning ability, some researchers propose Rejection sampling Fine-Tuning (RFT) (Yuan et al. 2023), which generates reasoning paths using LLMs and filters out those leading to incorrect answers. Moreover, the Self-Taught Reasoner (STaR) (Zelikman et al. 2022) use the ground truth answer as a hint for LLMs to generate reasoning paths that lead to correct answers. However, this approach may also produce incorrect reasoning paths that happen to arrive at the correct answer. Similarly, Iterative Reasoning Preference Optimization (Pang et al. 2024) samples correct and incorrect CoTs to form preference pairs and

iteratively applies Direct Preference Optimization (DPO) (Rafailov et al. 2024). These training methods fall into the category of outcome-supervised fine-tuning, as they directly supervise the LLMs to produce correct final answers rather than correct reasoning paths.

Process supervision provides LLMs with more precise and fine-grained feedback. Lightman et al. (Lightman et al. 2023) find that their process reward models (PRM) perform significantly better than outcome reward models (ORM). Using a PRM, reasoning steps can be rewarded, enabling reinforcement learning to train LLMs to generate better CoTs. However, Lightman et al. rely on human annotators to label the reasoning steps, which is highly expensive, particularly for challenging math problems. To automatically label reasoning steps, some works employ (Wang et al. 2024a,b; Luo et al. 2024) Monte Carlo Sampling to estimate the "correctness" of the steps, showing that PRMs trained on their automatically labeled data can even outperform those trained on PRM800K (Lightman et al. 2023), a human-labeled dataset. Wang et al. (Wang et al. 2024a) further train LLMs with their PRMs using reinforcement learning, and find that it is better than training with ORMs. ReST-MCTS* (Zhang et al. 2024b) leverages Monte Carlo Tree Search (MCTS) (Kocsis and Szepesvári 2006; Browne et al. 2012) to annotate the process reward of each step for training a PRM, and use the PRM to guide MCTS in turn.

Some studies propose methods that do not require a PRM to provide process supervision to LLMs. Step-DPO (Lai et al. 2024) and Self-Explore (Hwang et al. 2024) identify the first incorrect step in a reasoning path through step-by-step verification, creating a pairwise dataset for subsequent preference learning. MCTS-DPO (Xie et al. 2024) utilizes MCTS to generate pairwise data for preference learning, with training and data generation performed iteratively. However, MCTS-DPO labels reasoning steps only as chosen or rejected, which does not accurately capture the quality of the steps, which can lead to suboptimal performance. Additionally, MCTS-DPO forms preference pairs using only the best and worst steps, discarding other steps that are potentially valuable.

In this paper, we study using MCTS to generate process supervision data. We apply MCTS at each step along the reasoning paths generated by an LLM, assigning a score that captures "relative correctness" to the sampled next steps.

Compared to binary preferences, the scores reflect the quality of the steps more accurately. The next steps with scores are integrated into a weighted negative log-likelihood loss function to train the LLM. We also iteratively generate training data and train the LLM, following MCTS-DPO. Our experimental results demonstrate that the proposed methods considerably improve the performance of LLMs on two mathematical reasoning datasets. Furthermore, the models trained on one dataset also exhibit improved performance on the other, showing the transferability of the enhanced reasoning ability.

## Related Work

A key technique for enhancing reasoning is Chain-of-Thought (CoT) prompting (Nye et al. 2021; Wei et al. 2022; Kojima et al. 2022; Wang et al. 2022), which encourages LLMs to think about problems step by step and generate reasoning chains. It is also found that reasoning chains with more steps are more likely to lead to correct answers (Fu et al. 2022), further highlighting the importance of step-by-step reasoning. Furthermore, tree search algorithms, such as MCTS, are integrated with LLMs during inference to search for correct reasoning paths, resulting in significant improvements in performance on reasoning tasks (Hao et al. 2023; Yao et al. 2024; Qi et al. 2024; Zhang et al. 2024a), but at the cost of a substantial increase in inference compute.

While the aforementioned studies enhance reasoning during inference, another research direction aims to instill reasoning ability into LLMs through training. Some studies train LLMs using responses generated by the models themselves (Yuan et al. 2023; Zelikman et al. 2022; Pang et al. 2024; Trung et al. 2024), employing supervised fine-tuning or reinforcement learning. Question synthesis has also been shown to be effective for generating training data (Yu et al. 2023; Liu et al. 2024; Lu et al. 2024; Li et al. 2024), where several data augmentation techniques are commonly applied.

Recently, an increasing number of studies have demonstrated that process supervision is more effective than outcome supervision, for training both reward models (Lightman et al. 2023; Wang et al. 2024a,b; Luo et al. 2024) and LLMs (Lai et al. 2024; Hwang et al. 2024; Xie et al. 2024; Wang et al. 2024a; Chen et al. 2024). Learned process reward models are usually used for reinforcement learning and for selecting the best reasoning path from a set of sampled paths. In this paper, we explore training LLMs with process supervision without relying on reward models, thereby avoiding the complexity and instability of reinforcement learning.

## Proposed Methods

Assume that we have a dataset of reasoning problems (e.g., mathematical problems) and their corresponding answers $\mathcal{P} = \{(x^i, y^i)\}_{i=1}^N$. Our goal is to enhance the reasoning ability of the target LLM with $\mathcal{P}$ without human annotation. In addition, we assume that we are not accessible to LLMs stronger than the target LLM, so that our methods can be applied to the strongest LLMs.

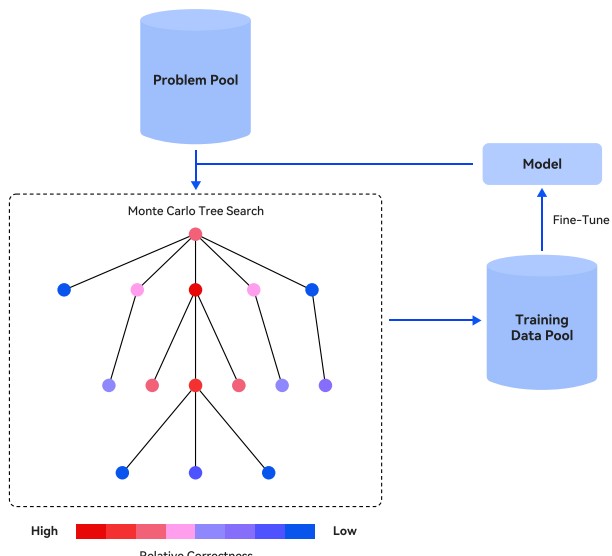

Figure 1: An overview of the proposed methods.

The proposed methods are illustrated in Figure 1. We train LLMs in a self-training manner (Zhang et al. 2024b; Zelikman et al. 2022), i.e., training LLMs on the data generated by themselves. We leverage MCTS to sample and search for step-by-step reasoning paths and collect training data from the constructed search tree. The generated training data are then used to perform supervised fine-tuning (SFT) on the LLM. This generate-and-fine-tune process is repeated iteratively until it converges.

In the following sections, we will describe in detail how we generate our training data and train the LLM on the generated data.

### Data Generation

We generate a training dataset $\mathcal{D} = \{(x^i, p_j^i, \mathbf{s}_j^i, \mathbf{c}_j^i)\}_{i=1}^N$, where $x^i$ denotes the $i$-th problem in $\mathcal{P}$, $p_j^i$ denotes the $j$-th partial solution to $x^i$, $\mathbf{s}_j^i$ denotes the next steps of the partial solution $p_j^i$, and $\mathbf{c}_j^i$ denotes the scores assigned to $\mathbf{s}_j^i$.

For each problem in $\mathcal{P}$, We regard each individual step in problem-solving steps as a tree node, and the steps are separated by two newline characters. We perform MCTS at each step along the reasoning paths generated by the LLM, exploring the best next steps. Specifically, for each partial solution $p_j^i$ of the problem $x^i$, we follow the iterative procedure below:

1. **Selection**. Starting from the root node (i.e., $p_j^i$), we select the child node with the highest Upper Confidence Bound (UCB) (Kocsis and Szepesvári 2006) value until the current node is either not fully expanded or represents a final step of the solution (e.g., "The final answer is 3.").

2. **Expansion**. If the current node does not represent a final step and has not been fully expanded, we sample the next

Table 1: Performance on MATH and GSM8K. Results are reported as mean $\pm$ standard error.

| Method | Llama-3.1-8B-Instruct | | deepseek-math-7b-instruct | |
| --- | --- | --- | --- | --- |
| | MATH | GSM8K | MATH | GSM8K |
| Zero-shot-CoT | $47.07 \pm 0.15$ | $80.77 \pm 0.25$ | $41.20 \pm 0.27$ | $78.79 \pm 0.20$ |
| RFT | $47.96 \pm 0.21$ | $83.33 \pm 0.25$ | $42.04 \pm 0.15$ | $80.90 \pm 0.18$ |
| Step-level DPO - Iteration 1 | $47.12 \pm 0.31$ | $82.43 \pm 0.27$ | $41.32 \pm 0.21$ | $80.67 \pm 0.25$ |
| Step-level DPO - Iteration 2 | $47.31 \pm 0.19$ | $82.55 \pm 0.17$ | $41.14 \pm 0.10$ | $80.52 \pm 0.23$ |
| Step-level DPO - Iteration 3 | $48.29 \pm 0.18$ | / | $41.48 \pm 0.30$ | / |
| Step-level DPO - Iteration 4 | $48.48 \pm 0.27$ | / | / | / |
| Ours - Iteration 1 | $50.04 \pm 0.07$ | $85.35 \pm 0.24$ | $43.15 \pm 0.28$ | $81.77 \pm 0.33$ |
| Ours - Iteration 2 | $50.84 \pm 0.20$ | $\mathbf{85.80 \pm 0.22}$ | $44.11 \pm 0.20$ | $\mathbf{82.02 \pm 0.18}$ |
| Ours - Iteration 3 | $51.52 \pm 0.18$ | / | $\mathbf{44.57 \pm 0.27}$ | / |
| Ours - Iteration 4 | $\mathbf{51.92 \pm 0.20}$ | / | / | / |

Table 2: Results of the transfer experiments where models are trained on one dataset and tested on the other.

| Method | Llama-3.1-8B-Instruct | | deepseek-math-7b-instruct | |
| --- | --- | --- | --- | --- |
| | GSM8K to MATH | MATH to GSM8K | GSM8K to MATH | MATH to GSM8K |
| Ours - Iteration 1 | $48.50 \pm 0.31$ | $85.21 \pm 0.15$ | $41.15 \pm 0.31$ | $80.36 \pm 0.38$ |
| Ours - Iteration 2 | $48.74 \pm 0.26$ | $85.72 \pm 0.14$ | $41.58 \pm 0.25$ | $81.09 \pm 0.18$ |
| Ours - Iteration 3 | / | $85.53 \pm 0.16$ | / | $81.10 \pm 0.17$ |
| Ours - Iteration 4 | / | $85.65 \pm 0.21$ | / | / |

step as a new child node using the LLM with a non-zero temperature.

3. **Simulation**. From the newly expanded node, we sample the continuation of the partial solution using the LLM until it produces a final answer.

4. **Backpropagation**. After the simulation, we compare the produced final answer with the ground truth, and propagate the reward (1.0 for correct and 0.0 for incorrect) back through the visited nodes in the tree, updating their visit counts and cumulative rewards to guide future searches.

After repeating the above procedure multiple times, we have constructed a tree where the child nodes of the root node represent the next steps of $p_j^i$. For the $k$-th node $v_{j,k}^i$ in the child nodes, its score is computed as follows:

$$r_{j,k}^i = \alpha \cdot N(v_{j,k}^i) \cdot \left( \frac{Q(v_{j,k}^i)}{N(v_{j,k}^i)} - \frac{\sum_m Q(v_{j,m}^i)}{\sum_m N(v_{j,m}^i)} \right) \quad (1)$$

where $Q(\cdot)$ is the cumulative reward of a node, $N(\cdot)$ is the visit counts of a node, and $\alpha$ is a manually set constant for controlling the scale of the scores. The expression in parentheses captures the "relative correctness" of $v_k$. Finally, we add 4-tuples $\{(x^i, p_j^i, s_{j,k}^i, r_{j,k}^i)\}$ into the training dataset, where $s_{j,k}^i$ is the step corresponding to $v_{j,k}^i$, except for the steps whose score is 0.

Then, we append the next step with the highest UCB value to $p_j^i$ to obtain a new partial solution, and perform the above procedure again, until the next step is a final step or the maximum solution length limit is reached.

## Iterative Training

We iteratively train the LLM after generating training data using the LLM from the last iteration, starting with the pretrained LLM at the first iteration. In each iteration, a certain number of problems are sampled from $\mathcal{P}$ for data generation.

At the $i$-th iteration, the LLM is trained by minimizing the following loss:

$$\mathcal{L}(\pi_{\theta_i}) = -\mathbb{E}_{(x,p,s,r)\sim\mathcal{D}_i}[r \log \pi_{\theta_i}(s \mid x, p))] +$$
$$\mathbb{D}_{KL}(\pi_{\theta_i}(s \mid x, p) \parallel \pi_{\theta_{i-1}}(s \mid x, p)) \quad (2)$$

where $\pi_{\theta_i}$ denotes the LLM at the $i$-th iteration. The first term is weighted negative log-likelihood. Inspired by reinforcement learning from human feedback (Ouyang et al. 2022), we incorporate a KL penalty in the second term to mitigate the distribution shift, which is a challenge in offline reinforcement learning. In fact, our training method can be regarded as a form of reinforcement learning.

# Experiments

## Setup

We apply the proposed methods to Llama-3.1-8B-Instruct[1] and deepseek-math-7b-instruct[2], and evaluate the performance on two popular mathematics datasets, GSM8K (Cobbe et al. 2021) and MATH (Hendrycks et al. 2021).

---

[1] https://huggingface.co/meta-llama/Llama-3.1-8B-Instruct
[2] https://huggingface.co/deepseek-ai/deepseek-math-7b-instruct

For comparison, the first baseline is Zero-shot-CoT (Kojima et al. 2022). The second baseline is Rejective Sampling Fine-Tuning (RFT) (Yuan et al. 2023), which generates training data by sampling the correct solutions generated by the LLM itself.

To demonstrate that our data generation method is superior to selecting the best and worst steps (i.e., the steps with the highest and lowest average rewards, respectively, similar to the method in (Xie et al. 2024)) to form preference pairs, we also iteratively train LLMs on the perference data generated in this way. We use DPO for preference optimization, and this baseline is referred to as Step-level DPO.

For efficient training, all models are trained using Low-Rank Adaptation (LoRA) (Hu et al. 2021) .

## Main Results

The experimental results are presented in Table 1. We report the results of iterative training until the accuracies do not increase anymore. It can be observed that our methods consistently outperform the baselines by large margins, with accuracies improving during iterative training. However, the performance converges quickly and fails to continually improve over more iterations. As a result, not all the problems in the training sets of MATH and GSM8K are used for training: only about 2,000 in MATH and 2,000 in GSM8K are used. In addition, the performance converges more quickly on GSM8K than on MATH, which we attribute to the lower difficulty of GSM8K, making it easier for models to learn. It is noteworthy that Step-level DPO achieves only very marginal improvements on MATH, which indicates that our data generation method is significantly superior.

## Transferability Evaluation

The MATH dataset consists of high school math competition problems, while GSM8K comprises grade school math problems. If the LLMs' mathematical reasoning ability has been improved, they should perform better on both datasets. Therefore, we evaluate the transferability of the models by training it on MATH/GSM8K and testing it on GSM8K/MATH to verify whether their mathematical reasoning ability has indeed been enhanced.

As shown in Table 2, our methods outperform Zero-shot-CoT on unseen datasets, indicating they indeed learn mathematical reasoning ability. As expected, the improvements are less substantial than those observed in non-transfer experiments, since the problems in the two datasets require different mathematical skills and knowledge.

## Conclusion and Limitations

In this work, we propose a process supervision data generation method utilizing MCTS and a training approach, for improving the reasoning ability of LLMs. We evaluate the proposed methods on two well-known mathematical datasets and demonstrating the effectiveness. The trained models also outperform the pre-trained models on datasets they are not trained on, showing the transferability of their learned knowledge and skills.

However, the performance converges quickly during iterative training, failing to continually improve over many iterations, and we do not even use all the problems in the training sets. Training for more iterations or using more problems not only fails to improve the performance but actually degrades it. Future research could study the underlying reasons for this phenomenon and how to achieve more substantial improvements with more iterations.

In addition, the models in our experiments are trained using LoRA, which could have limited the magnitude of the observed improvements.

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
