# OpenReview forum: "Enhancing Reasoning through Process Supervision with Monte Carlo Tree Search"
_AAAI.org/2025/Workshop/NeurMAD — AAAI 2025 Workshop NeurMAD Submission_

### Official Review · Reviewer_PeQ7 · 2024-12-20
**The paper proposes a MCTS based approach to generating process supervision data which is then used in training LLM (Llama 3.1 and DeepSeekMath).**

**Rating:** 6
**Confidence:** 5

**Review:**

The paper explores MCTS-based approach for automatically generating process supervision data. It would be better if a few things are clarified:

(1) What does 'm' stand for in Equation 1?

(2) Why wasn't the proposed approach compared to other MCTS-based approaches like:

Luo, Liangchen, et al. "Improve Mathematical Reasoning in Language Models by Automated Process Supervision." arXiv preprint arXiv:2406.06592 (2024).
Wang, P.; Li, L.; Shao, Z.; Xu, R.; Dai, D.; Li, Y.; Chen, D.; Wu, Y.; and Sui, Z. 2024a. , Math-shepherd: Verify and reinforce llms step-by-step without human annotations. (comparison with this will help evaluate the efficacy of the reward score function proposed)

(3) Process Supervision data size - what is the size of data used for training? How many problems from Math or GSM 8k was used? I suppose two sets of training data is used for each dataset - Math and GSM-8K.

---

### Official Review · Reviewer_Ezd1 · 2024-12-25
**Good Paper, Could be enhanced by benchmarking against prior research using MCTS**

**Rating:** 7
**Confidence:** 4

**Review:**

## Summary
---
This paper introduces a method that enhances reasoning abilities in large language models (LLMs) by integrating Monte Carlo Tree Search (MCTS) into process supervision. Unlike traditional approaches that focus on the final answer, this method scores intermediate reasoning steps based on their correctness, allowing for more nuanced training. Experiments on mathematical reasoning datasets (MATH and GSM8K) show significant improvements over baselines, with strong generalization to unseen datasets.


## Strengths
---
- Innovative use of MCTS to generate fine-grained supervision for intermediate reasoning steps, addressing limitations of outcome-based training.
- Consistent and substantial performance improvements are demonstrated on both in-domain and transfer tasks, supported by rigorous evaluation and clear comparisons to baseline methods like Zero-shot CoT and RFT, effectively validating the approach.

## Suggestions for Improvement
---
- Overstated Novelty: While the combination of MCTS and process supervision is novel, the paper could more explicitly acknowledge existing foundational work in these areas. Highlighting prior research on MCTS in reasoning tasks (e.g., its use in planning and decision-making algorithms) and process supervision methods would provide better context and strengthen the positioning of the contribution. Additionally, discussing how this work extends or diverges from existing approaches would clarify its unique value.

---

### Official Review · Reviewer_c4F6 · 2024-12-27
**Good Paper, Could be improved with few clarifications.**

**Rating:** 7
**Confidence:** 4

**Review:**

### Summary
-----
The paper leverages Monte Carlo Tree Search (MCTS) to generate process supervision data for enhancing the step-by-step reasoning capabilities of large language models (LLMs). Improving reasoning mechanisms in LLMs has been a longstanding challenge. Process supervision has shown better performance than methods focusing on producing correct outcomes. This work aims to train LLMs without relying on reward models, which are inherently complex, by augmenting data generated by the model itself. The proposed method samples and collects data from the search tree using MCTS, then performs supervised fine-tuning (SFT) on the LLM until convergence by minimizing the log-likelihood of the relative correctness of reasoning steps. The paper is well-written and easy to understand.


### Strengths
-----
- The experiments are comprehensively conducted with replications, standard error reporting, and notable performance improvements compared to baseline methods.
- The paper introduces a novel approach by suggesting data augmentation methods rather than relying on reward-based approaches, enabling greater training efficiency.
- The transferability evaluation demonstrates the model's ability to generalize effectively to unseen data.

### Suggestions for Improvements
-----
I kindly suggest clarifying the following points to improve the paper:
- Could the notation for the tree $\{(x^i, p_j^i, s_{j,k}^i, r_{j,k}^i)\}$ be embedded into Figure 1 for better visual alignment and clarity?
- Is this method the first approach to tackle data augmentation within the process supervision paradigm? If not, could references to related works be provided?
- What are the possible cases of distribution shift (e.g., label shift, covariate shift, domain shift) in Eq. (2), and can such shifts be minimized by the proposed method? Alternatively, does the term serve as a penalty term just to account for distribution shift?
- What could be the possible reason for the quick convergence observed? Could the use of self-generated data be a contributing factor?
- A more detailed explanation of the MCTS method itself would enhance readers' understanding. While the introduction mentions its use for annotation in previous works, the application in the proposed methodology appears to differ. Additionally, what are the strengths of MCTS compared to other baseline tools (if any exist) leveraged in handling reasoning process?

---

### Decision · Program_Chairs · 2024-12-30

**Decision:**

Accept

**Comment:**

 This is an interesting paper in reasoning. We agree with the opinions of reviewers.